# The Dual Role of the Airway Epithelium in Asthma: Active Barrier and Regulator of Inflammation

**DOI:** 10.3390/cells12182208

**Published:** 2023-09-05

**Authors:** Andreas Frey, Lars P. Lunding, Michael Wegmann

**Affiliations:** 1Division of Mucosal Immunology and Diagnostics, Research Center Borstel, 23845 Borstel, Germany; afrey@fz-borstel.de; 2Airway Research Center North (ARCN), German Center for Lung Research (DZL), 22927 Großhansdorf, Germany; llunding@fz-borstel.de; 3Division of Lung Immunology, Research Center Borstel, 23845 Borstel, Germany

**Keywords:** asthma, epithelium, inflammation, barrier, mucus, trained immunity, cytokines, IgA

## Abstract

Chronic airway inflammation is the cornerstone on which bronchial asthma arises, and in turn, chronic inflammation arises from a complex interplay between environmental factors such as allergens and pathogens and immune cells as well as structural cells constituting the airway mucosa. Airway epithelial cells (AECs) are at the center of these processes. On the one hand, they represent the borderline separating the body from its environment in order to keep inner homeostasis. The airway epithelium forms a multi-tiered, self-cleaning barrier that involves an unstirred, discontinuous mucous layer, the dense and rigid mesh of the glycocalyx, and the cellular layer itself, consisting of multiple, densely interconnected cell types. On the other hand, the airway epithelium represents an immunologically highly active tissue once its barrier has been penetrated: AECs play a pivotal role in releasing protective immunoglobulin A. They express a broad spectrum of pattern recognition receptors, enabling them to react to environmental stressors that overcome the mucosal barrier. By releasing alarmins—proinflammatory and regulatory cytokines—AECs play an active role in the formation, strategic orientation, and control of the subsequent defense reaction. Consequently, the airway epithelium is of vital importance to chronic inflammatory diseases, such as asthma.

## 1. Introduction

The World Health Organization (WHO) ranks chronic inflammatory diseases as among the greatest threats to human health since they represent the most significant cause of death worldwide and since their prevalence is still increasing. Among those chronic inflammatory diseases, bronchial asthma is the most common disease of the lung and affects more than 260 million people [1]. It is characterized by acute episodes of reversible broncho-obstruction and a number of symptoms, including chest tightness, coughing, shortness of breath, and wheezing [2]. Each of these symptoms may individually vary in duration, frequency, severity, and combination of occurrence. However, what they all have in common is that they arise on the basis of a chronic inflammatory response at the mucosal interface of the airways [3].

In principle, chronic inflammation can be regarded as an overreaction or permanent activation of the airway defense machinery. In healthy individuals, the defense system reacts to environmental stressors (e.g., pathogens) that harm or breach the epithelial boundary with a fine-tuned and adequate defensive response, and these reactions are downregulated once the offender has been taken care of. In asthma patients, however, mucosal homeostasis may become disturbed after a single trigger exposure and drift into the chronic inflammatory state upon repeated hits. Under certain circumstances, the latent inflammation aggravates suddenly, resulting in a perilous asthma exacerbation or status asthmaticus [4]. Amongst inducers of asthma attacks are exposure to pollutants and chemicals, tobacco smoking, and respiratory infections, but also psychological and environmental factors. As for disease initiation, it is important to note that asthma is not a single, homogeneous disease but rather a heterogeneous syndrome manifesting in different phenotypes, which are as yet not fully understood and characterized. For the development of the majority of—but not all—cases, current concepts of asthma pathogenesis propose a strong involvement of allergens. The allergen breaks through the barrier of the airway epithelium at some point in time and initiates an acute inflammatory response in the airway mucosa; subsequently, the inflammation may be reactivated upon further contact with an allergen. This subtype/endotype is called allergic asthma, whereby shifts and drifts between different allergic states are observed. Allergic asthma may start out from allergic reactions in the upper airways, such as hay fever, or it may develop from atopic dermatitis, where the main sites of inflammation switch from skin to airway mucosa, a phenomenon called atopic march [5,6]. Once established, allergic spectrum disorders usually bias T helper cell differentiation towards the TH2 phenotype, and the latent inflammation that persists between acute episodes directs immune cells (e.g., eosinophils, neutrophils, TH2 lymphocytes, etc.) to stay at or migrate to inflamed sites, which results in a continuous stimulus of immune cell-derived proinflammatory mediators, such as certain cytokines or chemokines. This, in turn, fuels a vicious cycle of tissue repair and destruction, causing airway remodeling along with constant airway hyperresponsiveness (AHR) and persistent mucus hyperproduction [7,8,9].

In contrast to the comparatively well-defined TH2-associated asthma phenotypes, a subset of asthma cases lacks typical type 2 inflammation and cannot be assigned any specific biomarkers. Identification of this non-T2 asthma relies on typical clinical features, such as airway obstruction and hyperresponsiveness, as well as sputum analysis. This phenotype is usually linked to strong neutrophilic airway inflammation [10,11].

Thus, no matter what asthma phenotype is present, the asthmatic airway mucosa stands in the crossfire of constant stimulus by immune cells from the inside and is challenged by microbes, noxae, allergens, and other foreign matter from the outside. Yet, the airway epithelium is not just a passive victim of this double-sided attack awaiting assaults hailing down on it. Rather, it actively carries out a broad spectrum of immune functions and is critically involved in the initiation, regulation, and conductance of immune responses, both towards the lumen and the stroma.

In the following, we want to outline how the airway epithelial layer is equipped to deal with this awkward position and what the short- and long-term effects may be for the epithelium itself, the neighboring stroma and lumen and their inhabitants, and lastly, for the entire lung and its afferent bronchi. We will put special emphasis on the function of the epithelial lining, the physiology of mucus, the role of the glycocalyx, the contribution of airway epithelial cells (AECs) to the development of secretory/protective humoral immunity, the role of AECs as sentinels of the environment, its regulatory effects on airway inflammation, and its ability to “remember” pathogen contact that impacts future reactions of the mucosa.

## 2. Cellular Architecture of the Airway Epithelial Lining

Although the airway is home to numerous cell types, the airway epithelium seems to be a key player in the pathological events associated with chronic airway inflammation. The epithelial layer is not a uniform entity of structurally and functionally identical cells but rather divides its tasks among different epithelial cell types. Some of those AECs constitute the outer frontier of the airway, while others reside under that boundary but are still located on the outer side of the basal membrane, which provides the structural substratum for the epithelium. For that reason, the epithelium is dubbed pseudostratified as the epithelial cells partially reside on each other, causing the nuclei to be on different levels, but each cell still maintains contact with the basal membrane. Along the naso-alveolar axis of the airways, various epithelial cell types constitute the cellular boundary to the environment. Their exact composition is difficult to define and appears to vary upon condition (e.g., healthy versus airway inflammation) and between individuals. Histological analyses can quantify fairly abundant cell types rather reliably, but even here, different data exist [12,13,14,15,16]; recent sophisticated analytical technologies, such as single-cell RNA sequencing, have allowed the identification of discrete subsets of epithelial cells and have established the presence of additional, rarer cell types in the human bronchi and bronchioles [17,18,19,20,21]. Yet, while these methodologies are highly sensitive and provide a plethora of particularized data, they usually reflect special situations that can identify specific alterations, e.g., in certain disease states [22,23,24,25], but make it difficult to obtain an overall picture [26]. In general, it can be stated that ciliated cells, goblet cells (GCs), and club cells (Clara cells) constitute the main differentiated epithelial cell types, with a change in frequency along the trachea-bronchial axis (Table 1). In addition, basal cells, which are not in contact with the lumen, form a loose mesh, allowing the boundary layer cells to extend to the substratum. The basal cells are key players in terms of biological memory because they are the principal stem cells and, as such, responsible for replenishing the epithelial layer after injury or in the course of physiological renewal [27,28,29]. Moreover, they can sense inflammatory changes in their microenvironment and adapt proliferation and differentiation accordingly [19]. It has been controversial for some time as to whether tuft cells (also termed brush cells), which have been unequivocally identified in the airways of rodents and other mammals, are also present in the human bronchi [27,30], but meanwhile, it could be demonstrated that human airways also harbor low numbers of tuft cells [19,20,31]. Functionally, tuft cells are believed to act as chemosensors, responding to environmental signals, e.g., by changing their surface receptor expression pattern. Upon activation, they can release neurotransmitters and various mediators involved in type 2 inflammation [19,32]. Pulmonary neuroendocrine cells are found only sparsely [18,20,33,34], although here, too, conflicting data exist [35]. They occur either as single cells or in small cell clusters, which are often in contact with nerve fibers [35,36,37]. They contain serotonin and a number of bioactive peptides stored in dense-core neurosecretory granules, and animal models of asthma suggest that they are involved in functions of the innate immune system as well as in adaptive immune responses [38,39,40]. In addition, their potential to induce goblet cell hyperplasia via the neurotransmitter γ-amino butyric acid indicates involvement in allergic asthma responses [41,42]. The recently identified pulmonary ionocytes [43] express high levels of the chloride transporter CFTR and are thought to play a critical role in airway surface liquid regulation. A schematic representation of the presence and distribution of the different airway epithelial cells along the bronchi and bronchioles is depicted in Figure 1.

**Table 1 cells-12-02208-t001:** Cells (as a percentage of epithelial cells) in the airways of healthy humans.

Cell Type	Bronchi	Larger Bronchioles	Smaller Bronchioles	Reference
Basal	27–30	27–30	4–23	Crystal 2014 [16] Boers 1998 [13] Walters 2014 [44]
		8 → 50	2–22	Okuda 2021 [15] Zuo 2020 [45] Deprez 2020 [20] Braga 2019 [18] Travaglini 2020 [17]
Ciliated	53	62–71		Staudt 2014 [46] Walters 2014 [44]
		5 → 50	30–60	Okuda 2021 [47] Zuo 2020 [45] Deprez 2020 [20] Braga 2019 [18] Travaglini 2020 [17] Paranjapye 2022 [48]
Goblet	11	1–10	0–2	Lumsden 1984 [14] Boers 1999 [15]
Club Cell (“Secretory Cell”)	0	1–30	10–40	Lumsden 1984 [14] Boers 1999 [15] Okuda 2021 [47] Zuo 2020 [45] Deprez 2020 [20] Braga 2019 [18] Travaglini 2020 [17] Paranjapye 2022 [48]
Neuroendocrine	“rare” (0.2–0.5)			Boers 1996 [34] Knight 2003 [33]
	“abundant” (2.5–10)			Weichselbaum 2005 [35]
Ionocytes	“rare” (<1–2)			Plasschaert 2018 [43] Braga 2019 [18] Scudieri 2020 [24] Travaglini 2020 [17] Paranjapye 2022 [48]
Tuft Cell (Brush Cell)	“rare”			Goldfarbmuren 2020 [49] Ualiyeva 2020 [31] Deprez 2020 [20]

After the onset of asthma, the architecture and composition of the airway epithelium changes over time and along with the severity of the disease—a process that also involves stromal aberrations, such as thickening of the smooth muscle cell layer—and has been termed airway remodeling (for review, see, e.g., [50,51]). The most obvious change in epithelial layer composition is the number of GCs, which can more than double in patients with severe asthma compared to healthy bronchi (goblet cell hyperplasia) [52,53,54]. Also, while virtually no dead cells are found in the airway epithelium in healthy individuals and people with mild or moderate asthma, a substantial number of dead cells are observed in the airway epithelium in the case of severe asthma, and the fraction of epithelial cells that extend from the layer increases drastically over disease progression [55,56]. The number of ciliated cells, on the other hand, declines from the healthy state over the mild and moderately asthmatic airway to the airways of persons suffering from severe asthma, while the quota of cilial losses and cilial misorientation on the remaining cells increases from the healthy to the severely diseased state [57]. Moreover, in rodent models of airway remodeling, increased collagen deposition between epithelial cells was observed, and while the number of mitochondria in the airway epithelial cells increased in the diseased state, those mitochondria lost their regular structure [58,59]. Thus, in particular, for severe asthma, architectural and structural changes in the airway epithelial layer are obvious, and the function of the airway epithelium is severely impaired.

## 3. Structure of the Luminal Boundary

The differentiated epithelial cells that constitute the boundary to the airway lumen form a tight, gapless monolayer, in which cell–cell contacts are sealed. Consequently, these cells are polarized, with a distinctly different architecture at the apical and at the basolateral side. The apical surface of the different cells involved in the epithelial barrier displays a variety of features (Figure 2). Their main function is to prevent or control access of foreign matter to and its translocation across the epithelium. In the upper airways, the dominant hallmark is a continuous blanket of mucus. It is produced by GCs and moved by ciliated cells, the two most prevalent epithelial cell types in this part of the bronchial mucosa. Those GCs directly located on the luminal surface mostly produce the highly glycosylated mucoproteins MUC5AC and MUC5B [60,61,62]. MUC5B prevails in the healthy mucosa because a major fraction of MUC5B is additionally provided by GCs in the submucosal gland acini [60]. In healthy individuals, mucus contains approximately 1% of salts, with Na^+^ and Cl^−^ reduced and K^+^ strongly increased in comparison to plasma and ≥95% water [63]. Upon release towards the lumen, the secreted mucus forms a sheet-like layer that covers the epithelium and is moved upward towards the larynx by an orchestrated, well-coordinated “beating” of the cilia. As the tips of the cilia just touch the mucus layer and propel it forward by frictional interactions [64,65], the mucus’ physical properties are critical for its transport. If the layer is too thick, it can occur that only the innermost portion is moved by the beating cilia, but the outer part remains in place. In order to be moved properly, it is also essential that the mucus layer is neither too viscous nor too runny. The proper fluidity is achieved by a fine-tuned water content and pH, which in turn is controlled osmotically via the electrolyte content of the aqueous phase. Animal models revealed that lack of electrolytes causes the mucus to stiffen and no longer respond to the beating cilia [66,67]. This is, for instance, the case in cystic fibrosis, where electrolyte transport is faulty due to a genetic defect in a chloride and bicarbonate channel (CFTR), as shown in human ex vivo models [68,69]. Another detrimental factor can be the failure of and the erroneous beating by some ciliated cells. This phenomenon, called secondary cilial dyskinesia, can result from misorientation and/or misdifferentiation of renewed ciliated cells after the repair of injuries in the outer epithelial lining. From this, vortexes may arise in the mucus flow, which slows down mucus transport and impairs the mucociliary clearance of foreign matter. Respiratory infections can have a similar effect, as bacterial toxins were found to also slow down cilial beating [70,71,72]. For efficient beating, the cilia are engulfed by a watery film of low viscosity, which is 7–10 µm in height and called the periciliary layer (PCL) [64,73,74]. As for mucus, the osmolarity and electrolyte content of this fluid sheet is extremely important. It was shown that PCL height is controlled by Na^+^- and Cl^−^-influx and efflux, which in turn is regulated by the adenosine triphosphate/adenosine ratio in the fluid [74], as well as by additional factors.

In the airways of both patients with asthma and animals with experimental asthma, the number of mucus-producing cells and the amount of mucin release is increased considerably, and the ratio of the different mucin changes. Though reports for the latter are not fully consistent, an overabundance of MUC5AC appears to be a common feature of asthmatic mucus [52,75,76,77]. Moreover, the physical properties of the mucus and the PCL are altered: mucin content in the mucus gel rises from 1–2% in healthy individuals to 8–15% in asthma patients [53,64], and the electrolyte composition in the PCL is changed [78]. Due to its higher protein content, asthma mucus attracts more water for solvation, which in turn depletes the PCL of water. With the watery phase of the PCL shrinking in height, the cilia extend further from the PCL and become bent (Figure 2) [64]. Both effects slow down mucus transport. On top of that, the mucin molecules may be oxidized or fail to be broken down proteolytically—as is the case in the healthy airway—which in turn results in a stiff mucus gel [79,80]. To make matters worse, the depletion of ciliated cells in the airways of patients with asthma, along with an increased occurrence of faulty or missing cilia, impairs the transportation machinery [57]. As a result, the asthmatic airway has to deal not only with cargo, which is more difficult to transport, but also with lower transportation performance. Unfortunately, the lack of functional ciliated cells cannot be compensated, e.g., by higher transportation rates of those that remain intact; on the contrary, the TH2-biased environment in the airways of asthma patients is a rich source of the cytokine interleukin-13 (IL-13), which is not only the main driver of GC-differentiation and mucus production but was also found to further induce aberrant and reduced cilial beating [81,82]. As a countermeasure to this predicament, the bronchial cilial beating frequency may be increased by serotonin administration and by agonists of the muscarinic receptor M3, as has been shown in animal models [83,84], but, though promising, these measures are still in the preclinical stage. Taken together, in patients with asthma, we see a coincidence of lower mucus transportability along with lower transport activity and, as a result, mucus plugging, which leads to an often fatal obstruction of the airways [85,86,87].

Beneath the mucus sheet and the periciliary layer, in which the cilia beat, one finds the outermost component of the epithelial lining cells, a macromolecular mesh consisting of membrane-tethered glycoproteins, such as the membrane-spanning mucins MUC1 and MUC4 and heparan sulfate [61,64]. Such a dense meshwork of cell membrane-anchored glycoproteins and glycolipids is called glycocalyx [88,89]. Although there may be a certain dynamic in this layer, due to some degree of lateral diffusion of the cell membrane constituents, the glycocalyx is much denser and more rigid than the mucus layer. It, therefore, provides a sub-micrometer thick unstirred fleece, whose porosity and height control the advance of foreign matter and the kinetics of this motion. The pore size of the airway glycocalyx was determined to be ≤40 nm [64]. While small proteinaceous matter, such as allergen molecules, may readily advance to the lipid bilayer of the cell membrane, larger particles, in particular viruses, which require direct contact with the lipid bilayer in order to infect cells, may not proceed if they exceed the pore size. In line with that, experiments conducted on the intestinal epithelium show that only nanometer-sized particles are able to advance to the surface of epithelial cells [90,91,92,93]. More recent work confirms these findings for the airways, demonstrating that 30 nm virus particles can penetrate the glycocalyx of ciliated cells, whereas virus particles of 100 nm in diameter cannot reach the cell surface [61]. Consequently, evolutionary optimized adeno-associated viruses (AAVs), which are approximately 20–25 nm in diameter and use constituents of the glycocalyx (heparan sulfate in the case of AAV2) as receptors, function well in airway epithelium gene therapy [94,95]. On the other hand, larger-sized adenoviruses (90–100 nm diameter [96]) whose receptor resides on the basolateral surface of the epithelium [97] often perform poorly as transfectants for airway epithelial cells [76]. Usually, engineering of viral vectors is necessary [98,99], and it remains the subject of debate whether the airway surface must be prepared such that the physical barrier becomes disrupted in order to achieve effective viral-mediated lung gene therapy [100]. Thus, the physical hurdles on top of the airway epithelium seem to be well-suited to prevent infection. In the case of asthma, however, the increased mucin content of the airway mucus attracts water and is thus able to diminish the height of the periciliary layer [66]. If this happens, the macromolecular mesh consisting of membrane-tethered glycoproteins is stripped downwards into a compressed stage at the bottom of the cilia, and the tips of the cilia extend towards the lumen into the mucus without any access-blocking glycocalyx coat on them (see Figure 2) [66]. Consequently, those cilia may become vulnerable to viral infection as long as the viral receptor resides in the membrane of the cilia. Although not proven experimentally and, thus, still hypothetical, such a situation may lead to more frequent viral airway infections in asthma patients, often observed (e.g., back-to-school asthma) [101], and a potential cause for exacerbations.

The main hurdle at the next inner tier is tight junctions. Tight junctions are molecular gaskets that prevent paracellular leakage between cells. Within the cell, they are held in place by the cytoskeleton and run around the entire cell, merging with the same structure of all neighboring cells. The tightness of the resulting sealing is reflected in the high electrical resistance that a polarized epithelial layer provides in cell culture [102,103]. While not directly demonstrated for the airway epithelium, in vivo studies on the tight junction-sealed gut epithelium show that even during epithelial turnover, tightness of the layer is maintained [104,105]. As tightness is an important criterium for the healthy airway epithelial lining, it seems within reason that a similar mechanism to gaplessly remove attrite cells operates in the airways, too.

For the airway of patients with asthma, it was demonstrated that tight junction integrity is impaired [106], possibly caused by activation of the mechanosensitive receptor piezo-1 upon the higher airway pressure present in these patients [107]. Besides mechanical effects from the luminal site, infection with certain airborne pathogens, such as human rhinovirus (hRV), were shown to compromise epithelial barrier integrity and facilitate the transmigration of bacteria across primary human AEC layers in culture [108]. In addition, proteases derived from different plant pollens have been described to degrade intracellular junction proteins and increase transepithelial permeability [109,110]. Although tight junction impairment in biopsies and cultured primary human and murine AECs seems to occur already without any signaling from the stromal side, basolaterally applied IL-13 and, to some extent, IL-4 was able to boost paracellular permeability [111]. Conversely, epidermal growth factor applied onto biopsies from the apical site is able to stimulate tight junction formation, thereby restoring barrier function to a large extent [106].

## 4. Molecular Function of the Boundary and the Epithelial Cells Constituting It

The mucus is the outermost interface between the epithelium and the inhaled and exhaled air. It moistens the moving gas and traps foreign particulate matter. Besides this passive barrier function, mucus holds epithelium-borne antimicrobial peptides and proteins (e.g., lysozyme, lactoferrin), which are aimed at taking out bacterial pathogens [112,113,114]. These defensive mechanisms, however, can be compromised in their function in the case of mucus hypersecretion and/or overproduction of mucins. High concentrations of airway mucins—especially MUC5AC, which is elevated in patients with asthma—can interfere with the bactericidal activities of antimicrobial substances and can inhibit bacterial killing by neutrophils [115]. While this effect is of the highest importance in lung diseases, which are known to be associated with chronic bacterial infections, such as cystic fibrosis and COPD, it is also a critical aspect in asthma, where bacterial infections have been linked specifically to the exacerbated state, too [116,117].

A further class of defensive molecules that are contained in mucus are secreted antibodies of class A (sIgA) that have been released by the epithelium. In our canonical view of sIgA production, the first step is that luminal antigens are picked up and translocated across the epithelium by M cells, a specialized epithelial cell type [92,118], and by dendritic cells (DCs), which extend their dendrites towards the lumen without disrupting the tight-junction-sealed outermost epithelial layer [119,120,121]. Once the foreign matter is taken up, it is delivered to underlying organized mucosal lymphoid tissues (O-MALT), where IgA are produced by specialized B lymphocytes and assembled into dimers joined at their constant region via the J-chain (joining-chain) [122,123]. After release from the B lymphocyte, these dimeric IgA are picked up by poly Ig receptors (pIgR) at the basolateral side of columnar epithelial cells, the receptor-IgA complexes are translocated across the epithelial layer, and the IgA is released at the apical side, with a remnant of the poly Ig receptor, the secretory component (SC), bound to it [124,125]. From there, the sIgA diffuse into the mucus layer, where they can exert their protective and regulatory functions. Notably, the sIgA are already protective while travelling through the epithelial cells, as they can, e.g., intercept and neutralize viruses, which are in the course of being transcytosed across the epithelium [126].

As long as the sIgA have not encountered their respective antigen and have not become crosslinked via multivalent antigen binding, they seem to play an anti-inflammatory role. Eosinophils are not activated upon contact with non-loaded sIgA but are enhanced in their survival; basophils that have been stimulated with IL-3 will not release histamine or other inflammatory mediators upon contact with free, uncomplexed sIgA. However, once the sIgA is crosslinked and immobilized in a manner similar to its decorating the surface of a pathogen, the entire complex will activate basophils and eosinophils [127,128]. Apparently, the multi-valency of a sIgA-decorated pathogen surface, which provides multiple binding sites for IgA receptors, seems to be an important discriminator. A similar ambiguity as for sIgA was observed in the case of the antimicrobial protein lactoferrin and eosinophils; again, only the crosslinked molecule seems to activate the eosinophils [129]. However, sIgA-antigen complexes do not merely act as activators for eosinophils and basophils. They have also been found to bind to mucus [130] and, thus, may be removed by mucociliary clearance before any activation of basophils and eosinophils occurs.

As a class-switched and B-cell-derived immunoglobulin, secretory IgA used to be considered a member of the adaptive immune system. This dogma may no longer be valid to the full since, in recent years, natural antibodies have gained attention. These antibodies are non-affinity-matured immunoglobulins whose antigen-binding regions are inherited. According to our current definition, they would belong to the innate immune system. Natural antibodies mostly display paratopes, which are able to recognize common surface structures of microbial pathogens, such as glycoproteins, glycolipids, phospholipids, and oxidized variants thereof [131,132,133,134]. Natural antibodies that react with DNA have also been found [135]. As a kind of an “immunological starter kit,” they can be produced rapidly upon pathogen exposure and may fulfill the role of a pawn sacrifice until affinity-matured immunoglobulins against the respective intruder have evolved. Although discussed for approximately 50 years and discovered as gene products of B1a cells some 30 years ago, appreciation of those natural antibodies and their function as the first line of defense is rather new [134]. When assessing the mucosal immune defense, we must, therefore, now consider the naturally-produced mucosal IgA along with the T cell-independently produced mucosal IgA and the T cell-dependently-produced mucosal IgA as joint protectors of the mucosal lining.

While the protective role of sIgA in the gastrointestinal tract was proven directly some 20 years ago by using an in vivo model (“backpack tumor” model), where anti-pathogen dimeric IgA is produced and released in syngenic, immunologically naive mice [136,137,138,139], the role of sIgA on the mucosal lining of the airways is still subject of debate [140,141]. Studies analogous to the “backpack tumor” model have not been conducted for the lower airways as yet, but a protective role of “backpack tumor”-borne specific sIgA upon nasal challenge with Salmonella spp. has been published [142]. Nevertheless, our state of knowledge in the field is rudimentary compared to the work published on sIgA function in the alimentary tract, although work on sIgA function in the airways has gained momentum in the recent past.

Mice whose epithelial IgA transport machinery was knocked out genetically (pIgR −/−) spontaneously develop chronic airway inflammation. Furthermore, pIgR deficiency is associated with airway remodeling, emphysema along with enhanced matrix metalloprotease-12 and neutrophil elastase expression, an altered lung microbiome, and bacterial invasion of the airway epithelium as well as leukocyte infiltration. Under germ-free conditions, on the other hand, no signs of airway inflammation are being observed [143]. Increased epithelial leakiness, as found in the gut in pIgR −/− mice [144], could be one of the reasons why such airway pathology develops. Although the phenotype developing in the pIgR −/− mice more resembles that of a chronic obstructive pulmonary disease (COPD) in humans than that of asthma, it clearly demonstrates that the production and release of sIgA in the respiratory tract is indispensable for airway protection and crucial for the maintenance of airway homeostasis. This is not surprising as sIgA is believed to play a central role in the conditioning of the mucosal flora [138], though again, no studies on a possible interplay of sIgA with the lung microbiota have been conducted as yet.

In the case of asthma, direct proof for a protective role of luminal sIgA is still missing, but correlative relationships for sIgA deficiency and asthma or asthma aggravation have been published. Arguments against a beneficial role of sIgA in the respiratory system were found for stromal fibroblasts, which apparently proliferate and produce proinflammatory cytokines upon contact with sIgA but not with monomeric serum-borne IgA in cases of idiotypic pulmonary fibrosis [145]. The sIgA was found predominantly in the airspaces close to hyperplastic epithelia and implied to leak across the epithelium [146]. As hyperplastic and, thus, damaged epithelia are known to be leakier than a healthy epithelium, the activation of fibroblasts may also be a downstream effect of an already damaged epithelium that allows luminal sIgA to pass through toward the stroma. In the case of a healthy and, therefore, tight epithelium, the sIgA would not have access to stromal fibroblasts. The fibroblast may thus act as a leak detector in this situation. Regarding the activation of eosinophils by complexed sIgA, an adverse effect of this secreted immunoglobulin was proposed as well [128]. Yet, activation of eosinophils by complexed sIgA or lactoferrin may not necessarily lead to epithelial damage. This would happen only if the sIgA or the lactoferrin were complexed on the epithelial surface. If not, and if the sIgA decorates a microbial pathogen at some distance from the epithelial layer, the eosinophil attacks the pathogen that is bridged to it via the sIgA coat on the microbe.

Supporting a beneficial role of sIgA in chronic inflammatory airway diseases, such as asthma, are reports that the pIgR is downregulated in eosinophilic upper airway diseases [147] and that bronchial secretion of IgA is impaired in the airways of asthma patients due to the downregulation of pIgR [148]. Again, the notorious cytokines IL-4 and IL-13 seem to be responsible for this effect, which appears to be mediated by autocrine production of transforming growth factor beta (TGF-β) by the epithelium, as found in vitro on primary human AECs [148]. A similar suppressive effect of TGF-β was also observed for biopsies and primary cells of patients with chronic obstructive pulmonary disease (COPD) [149]. Yet, as is often the case, the role of cytokines can be ambiguous. In an in vitro model of human airway epithelium (adenocarcinoma cell line Calu 3), TGF-β was found to upregulate pIgR expression via p38 MAPK activation [150]. TGF-β expression was influenced by the redox balance as well as by the presence of neutrophils in coculture experiments. Here, the secretory leukocyte proteinase inhibitor SLPI further promoted pIgR expression [128]. Upregulation of pIgR in the airway epithelium can also be induced by IL-17 [151]. Hence, proper expression of pIgR seems to be at least as important as the availability of stromal dimeric IgA for humoral mucosal immune defense in the airway lumen. Overall, the expression of pIgR seems to be controlled in a fine-tuned manner by cytokines, which largely originate from leukocytes, cells residing underneath the epithelium, and to some extend by the AECs themselves. Thus, the epithelium is a key actor, but the play is directed, to a large extent, by interaction with immune cells residing on the stromal side of the airway lining.

## 5. AECs as Sentinels of the Environment

Located at the border between the body and its environment, AECs are involved in the protection of the integrity of the organism. Over the last few years, it became increasingly obvious that AECs are not only important for the maintenance of the structural barrier that separates the organism from its environment but also play an important role in the first line of immune defense as sentinels for external and internal stress factors including pathogens and other danger signals. These signals are classified as danger-associated molecular patterns (DAMP) and pathogen-associated molecular patterns (PAMP).

DAMPs are molecules that derive from injured or destroyed cells and act as clear signals indicating the loss of homeostatic integrity of specific cell compartments or even whole cells. DAMPs can originate from different compartments, such as the extracellular matrix (ECM) (fibronectin, hyaluronan, versican), mitochondria (ATP, mitochondrial DNA, N-formylated peptides), the nucleus (High-Mobility-Group-Protein (HMGB1)), the endoplasmic reticulum (calreticulin), or the cytoplasm (S100 proteins, heat shock proteins, defensins, galectins, uric acid) [152]. During necrosis, for instance, the DNA binding protein HMGB1 is released and can be sensed by nearby cells through the receptor AGE (RAGE). PAMPs are strongly preserved molecules and structures from pathogens and toxins. PAMPs originate from different sources, such as viruses, bacteria, fungi, and parasites [153]. After binding to their specific receptor, both DAMPs and PAMPs trigger their particular signaling pathways that lead to the activation of the immune system. Thus, ligation to and signal transduction by MyD88 and MyD88-independent mechanisms eventually cause activation of nuclear factor kappa b (NFkB), interferon regulatory factor (IRF) 3, or mitogen-activated protein (MAP) kinases [154,155,156,157]. This ultimately results in the production of proinflammatory mediators and, consequently, in the recruitment of immune cells. The type of the triggering DAMPs and PAMPs, along with the nature of the foregoing inflammatory environment, determines if the signals will promote protective or pathological effects for the host organism [158].

AECs, on the one hand, represent an important source of DAMPs, and on the other hand, they can also actively respond to these signals [159,160]. DAMP/PAMP recognition by pattern recognition receptors (PRR) can occur both extracellularly and intracellularly via mainly four different groups of receptors, namely toll-like receptors (TLR), C-type lectin receptors (CLR), Nucleotide-binding oligomerization domain (NOD)-like receptors (NLR), and retinoic acid-inducible gene-I-(RIG-I)-like receptors (RLR) [161]. The nature of PRR signaling in allergic inflammation is complex and multidirectional. In an experimental model of asthma, the administration of LPS before sensitization reduced the severity of the allergic process, while in the case of joint injection with an allergen, it increased the influx of macrophages, lymphocytes, and neutrophils into the lungs, aggravating the inflammatory phenotype [162].

TLRs are by far the best-studied PRRs. So far, 11 different TLRs have been identified in humans [163]. The structure of TLRs is composed of N-terminal leucine-rich repeats (LRRs) responsible for binding, a transmembrane region, and a cytoplasmatic Toll/IL-1R homology (TIR) domain responsible for signaling [164,165]. Over the last two decades, many TLR ligands have been identified: TLRs-1, -2, and -6 allow recognition of microbial lipoproteins [166,167,168,169]. TLR4 recognizes the cell wall component lipopolysaccharide (LPS) of gram-negative bacteria [170]. TLR5 senses flagellin present in large amounts on nearly all flagellated bacteria [171]. TLR9 recognizes unmethylated DNA sequences as evidence of the presence of bacterial DNA [172]. TLRs-3, -7, and -8 all recognize different forms of RNA that originate from viruses. TLR3 binds double-stranded RNA, whereas TLR7 and TLR8 bind GU-rich single-stranded RNA in different cell compartments, such as endosomes [173,174,175]. Though little is known about the ligands of TLR10, it appears to exhibit anti-inflammatory properties. Most recently, it was discovered that Pam3Cys, FSL-1, and HIV-gp41 could bind to and induce signaling via this receptor [176,177]. Finally, TLR11 recognizes uropathogenic E. coli and molecules from toxoplasma [178,179]. Whereas TLRs-1, -2, -4, -5, and -6 are located extracellularly in the plasma membrane, TLRs-3, -7, -8, and -9 are located intracellularly in the membrane of the endoplasmic reticulum (ER) and other cell compartments [180].

The plasma membrane-bound CLRs, originally characterized by a C-type lectin domain, consist of a large number of structurally different molecules [181]. Sub-classified into those that have an immunoreceptor tyrosine-based activation motif (ITAM) and those with an immunoreceptor tyrosine-based inhibition motif (ITIM), CLRs either have an activating or an inhibiting effect on the immune system, respectively. They are especially, but not exclusively, important for the recognition of fungal PAMPs and are involved in the recognition of apoptotic processes in their cellular neighborhood. Though they are expressed by AECs, much about their involvement in epithelial immune responses remains to be elucidated. At least, for dectin-1, it has been demonstrated to be involved in the detection of mycobacteria and *Aspergillus* spp. [182,183]. It was shown that β-glycan moieties within house dust mite (HDM) extract induce CCL20 in AECs [184] via CLRs and that secretion of TSLP by AECs increased carbohydrate-dependently upon stimulation with Der p 1 [185]. Lipoglycans of M. tuberculosis were strong inducers of hepcidin expression in AECs [186], and most recently, the expression of a functional mannose receptor in human bronchial epithelial cells was shown, all caused by CLRs [187].

NLRs are sensing PAMPs within the cytosol. Their structure is characterized by a central nucleotide-binding and oligomerization domain (NOD or NACHT) and C-terminal leucine-rich repeats (LRRs) as PAMP binding sites [188]. The NOD1 receptor senses gram-negative peptidoglycan [189,190], whereas the NOD2 receptor is a general sensor of bacterial peptidoglycan [191].

Finally, similar to TLR3, the RLRs RIG-I and melanoma differentiation-associated protein 5 (MDA5) represent cytosolic receptors recognizing genomic or intermediate dsRNA motifs of RNA viruses [192,193].

With the expression of this broad range of PRRs, AECs are perfectly equipped to fulfill the role of sentinels of the immune system [163,194]. Within the first two days of infection, the epithelial cells are considered the main source of anti-viral responses [195]. After sensing pathogens or other danger signals, AECs secrete antimicrobial peptides, enzymes, reactive oxygen species, and a plethora of different chemokines and cytokines to contribute to the initiation, regulation, and maintenance of immune responses, especially during the early phase of infection. Thereby, the signals of TLRs, CLRs, NLRs, RLRs, or RAGE cooperate to specifically regulate cellular immune responses to cell stress, infection, and inflammation, which can amplify or dampen their effects. Recurrent cell stress and repeated exposure to pathogens trigger chronic activation of PRR pathways in AECs, resulting in their highly active and important role in the pathogenesis of chronic inflammatory airway diseases [152].

There is evidence that this sentinel function is impaired in the airways of patients with asthma. Single nucleotide polymorphisms (SNP) in five different TLRs (TLR2, TLR4, TLR6, TLR9, TLR10) were described to significantly contribute to asthma susceptibility, severity, and responsiveness [196,197,198,199]. A polymorphism in TLR4, for example, was associated with a decreased forced expiratory volume in 1 s (FEV1), suggesting a reduced lung capacity and increased airway resistance [200]. Additionally, a polymorphism in the TLR2 gene was shown to be associated with the risk of asthma development [201], while another TLR2 polymorphism had a protecting effect [202]. Furthermore, a polymorphism in the TLR6 gene contributed to the development of asthma and allergic rhinitis, whereas an SNP in the TLR10 gene was negatively associated with asthma development [202]. These disease associations make TLR interesting as a possible target for therapeutic interventions. Therefore, several new drugs targeting different TLRs (TLR3, TLR4, TLR7, TLR8, TLR9) are currently under examination for patients with asthma [203,204,205,206,207,208,209,210,211] (Figure 3).

## 6. AECs as Regulators of Inflammation

Expressing a wide spectrum of PRRs (e.g., TLRs, NLRs, RAGE), AECs are equipped to respond to manifold signals that arise from cellular stress and destruction as well as from pathogen contact and infection. Activation of these PRRs initiates the expression and release of chemokines, cytokines, and growth factors by AECs, enabling them to regulate the recruitment, activation, and differentiation of T cells, B cells, dendritic cells, eosinophils, and neutrophils at inflammatory sites. Depending on the pattern by which PRRs are activated, AECs react with the production of certain mediators. Therefore, they are not only involved in the classification of the respective stressors but also contribute to the decision of how to react to them.

Bacterial infection leads to the activation of, e.g., TLR2 or TLR4, NOD1 or NOD1 expressed on AECs and subsequently to the release of proinflammatory cytokines, such as IL-1β, IL-6, and TNF, as well as of chemokines, such as CXCL8, CCL11, and CCL20. Together with the secretion of antimicrobial peptides, these mediators initiate an antibacterial immune response and contribute to bacterial clearance. In contrast, activation of MDA-5, RIG-I, TLR3, and/or TLR7/8 does not only induce expression of IL-1, IL-6, and TNF but also of type I interferons, such as IFN-α and IFN-β, and the type III IFN-λs, which provide an anti-viral environment for surrounding cells in order to diminish viral replication and to promote epithelial apoptosis. In cases of infection with either bacteria or viruses, AECs initiate a local inflammatory reaction that aims to counteract the invading pathogen and prevent its further spreading until the subsequent response of the adaptive immune system is established. However, in the airways of patients with asthma, the release of the abovementioned mediators by AECs may have a large impact on the already established allergic immune response in the airways and, thus, the course of the disease. The role of type I and III interferons in asthma pathogenesis has been comprehensively reviewed elsewhere, and it appears that deficiencies in these mediators can have considerable effects. In contrast, AEC-borne cytokines, such as IL-1, IL-6, and TNF, are not specifically tailored to direct immune responses against bacteria and viruses but also generally promote all types of inflammation and may be released by AECs of asthmatic airways without any contact with inhaled noxae [212]. Consequently, they also amplify the allergic inflammation of the airways, which leads to acute aggravation of the disease symptoms and necessitates an adjustment of the medication to regain symptom control. Such acute asthma exacerbations represent a major need for medical intervention, creating a substantial financial burden [213] generated by asthma and are the clearest indicator of the morbidity of this disease [214].

Since IL-1β expression has also been correlated with neutrophilic airway inflammation in patients with severe, steroid-resistant asthma, it has been suggested that it plays a significant role in the pathogenesis of this disease subtype [215]. As for the source of IL-1β, it has been shown that activation of the NLRP3 inflammasome by infection with either *Chlamydia muridarum* or *Haemophilus influenzae* and subsequent activation of caspase 1 leads to the enhanced release of IL-1β. IL-1β causes subsequent influx of neutrophils into the airways of mice predisposed by ovalbumin (OVA) induced experimental asthma that is refractory to dexamethasone treatment. In contrast, inhibition of either NLRP3 or caspase 1, as well as neutralization of IL-1, resulted in reduced airway neutrophilia and disease pathology. Remarkably, IL-1β administration restored the steroid-resistant features of the modeled disease [216]. In as much as the extent of the airway, neutrophilia is positively associated with both the severity of asthma and the degree of corticosteroid resistance; recruitment of neutrophils to asthmatic airways is suggested to be of critical importance to the pathogenesis of this asthma phenotype, which could also involve activated TH1 cells [217,218,219]. Concerning neutrophil recruitment, IL-17A-mediated mechanisms are suspected to primarily direct neutrophil migration to the airways of patients with severe asthma. The main source of IL-17A appears to be TH17 cells but not the airway epithelium [220,221,222]. However, these cells efficiently induce the release of IL-8 and CXCL-1, the most potent chemotaxins for neutrophils, and granulocyte-colony stimulating factor (G-CSF) in AECs in vitro by secreting IL-17A [223,224]. In turn, AECs of both humans and mice have been shown to be a major source of TARC/CCL17, the main chemoattractant for TH17 cells, in response to viral infection and/or stimulation with the TH2-type cytokine IL-4 [225,226,227]. This opens the possibility for a positive feedback loop, propelling a rapid influx of TH17 cells and neutrophils during viral airway infection of asthma patients. These findings illustrate the somehow Janus-faced functions of AECs in regulating immune responses in the lung: one the one hand, they play a significant role in the defense against invading pathogens by promoting acute (neutrophilic) inflammation. However, by performing this under the conditions of an already inflamed/diseased asthmatic airway—as described above, for example, in TH1 high and TH17 high inflammatory asthma subtypes—they also contribute to acute exacerbation and steroid resistance of the established disease.

Besides their role as sensors of bacterial and viral infection and their involvement in the recruitment of neutrophils, over the last decade, AECs have also been identified as central regulators of type 2 immunity. Especially by releasing factors such as IL-25, IL-33, and thymic stromal lymphopoietin (TSLP), AECs provide regulatory signals to cells of the innate as well as of the adaptive immune system not only during initial processes in response to allergen or parasite contact but throughout the entire inflammatory process. Since these three cytokines alert the immune system to external injury and subsequently initiate repair processes to restore destroyed tissues, they are also dubbed alarmins.

An increased release of TSLP, a member of the IL-2 cytokine family, as well as several genetic variants of it, has been recorded for a number of atopic diseases, including asthma [228]. TSLP is released by AECs during both inflammatory and homeostatic conditions [229], and the receptor for TSLP (TSLPR) is widely expressed among cells that are involved in the pathogenesis of bronchial asthma, including basophils, dendritic cells (DCs), innate lymphoid cells (ILCs), and T cells [230]. Several studies carried out in vitro in human and murine cells, as well as in vivo in mice, demonstrated its impact on the development of TH2-type immune responses. Hence, mice with T cells lacking the TSLPR fail to generate a TH2-type memory after immunization with OVA/alum [231,232,233]. In line with this, it has been shown in a number of mouse models of allergic diseases that TSLP can directly induce the differentiation of TH2 cells [234,235,236]. These findings can be explained by the direct effects of TSLP on DCs, T cells, and ILCs. Thus, stimulation of DCs would not only lead to the upregulation of costimulatory molecules, such as CD80, CD86, and OX40, but also induce the production of the TH2-type cytokines IL-4, IL-5, and IL-13 [231,232,233]. With that, TSLP-conditioned DCs have been shown to facilitate the maintenance of TH2-type memory effector cells [237]. TSLP is also able to directly promote TH2 cell differentiation and expression of TH2-type cytokines in naïve T helper cells, even independently of initial IL-4, if these cells receive stimulation via their TCR [235,238,239,240].

IL-25 (also known as IL-17E), a member of the IL-17 cytokine family, is produced by AECs as well as by TH2 cells, macrophages, mast cells, and eosinophils [241,242,243]. Patients with asthma display a constitutively high expression of IL-25 in the airway epithelium [243]. The receptor for IL-25 is constituted by IL-17RB and IL-17RA, which recruit the adaptor protein Akt1 and is expressed by immune cells, such as T cells, ILCs, and inducible natural killer (iNKT) cells, as well as by structural cells, including epithelial cells, fibroblasts, and endothelial cells [244]. Inhibition of IL-25 signaling, e.g., by an IL-25 neutralizing antibody, diminishes allergic airway inflammation, airway hyperresponsiveness, and airway remodeling in mouse models of experimental allergic asthma [245,246]. Another study could trace this effect back to IL-25, which stimulates an IL-17RB-expressing subset of iNKT cells to release TH2-type cytokines, such as IL-4, IL-5, and IL-13 [247]. In this context, IL-25 has further been shown to amplify TH2 cell differentiation in an IL-4-dependent fashion [248] and to elicit group 2 ILC formation that, in turn, also releases IL-4 [249,250].

IL-33, a member of the IL-1 cytokine family, is constitutively expressed by both epithelial and endothelial cells [251], and genetic variations of IL33 and its receptor IL-1RL1 (ST2) have been associated with asthma in humans [252,253,254,255,256]. In contrast to other members of its cytokine family, full-length IL-33 is biologically active; however, its activity can be modulated, while a number of allergen-, mast cell-, and neutrophil-derived proteases have been described to further activate IL-33 [257,258,259], cleavage by caspases-1, -3 or -7 lead to its inactivation [260,261,262]. The receptor for IL-33, formed by IL-1RL1 (ST2) and IL-1AcP, is constitutively expressed by TH2 cells, ILC2s, regulatory T cells (Tregs), and mast cells but can also be induced in other immune cells as well [263,264,265,266,267,268]. As shown in mouse models, IL-33 appears to particularly affect TH2 memory cells and ILC2s but in different ways. While memory TH2 cells, for example, reveal a higher expression of ST2 than effector TH2 cells and release amphiregulin in response to IL-33 [269,270], ILC2s are potently activated by IL-33, as shown by surface expression of OX40L and PD-L1, and they release considerable amounts of IL-5 and IL-13 [271,272,273,274]. In line with that, IL-33 was shown to be critical for the activation of ILC2s and TH2 memory cells in a mouse model of HDM or papain-induced experimental asthma [270,275,276,277].

Thus, under certain conditions, AECs act as potent inducers of TH2-type immune responses and, therefore, of the allergic reactions underlying the formation of experimental allergic asthma. That not being enough, the subsequent release of TH2-type cytokines by infiltrating TH2 cells and ILC2s, in turn, has considerable effects on the airway epithelium. In particular, IL-13 induces the expression of MUC5AC and, therefore, the differentiation of AECs into mucus-producing GCs [278]. De-regulated production and secretion of mucus that intrinsically is a part of the defense mechanism against airborne noxae leads to the formation of mucus hypersecretion, which is a prominent pathologic feature in asthma [279]. Furthermore, it has been demonstrated that stimulation with IL-4 polarizes AECs into an “E2-phenotype” that is characterized by expression of the transcription factors GATA-3 and STAT6, the receptors IL-1RL1 (ST2) and IL-4Rα, and of mediators, such as CCL26 (eotaxin-3), IL-24, and IL-33. This offers the possibility of a vicious-cycle-type feedback loop fostering a TH2-type local micro-milieu [280].

With the release of these cytokines, activated AECs contribute to the formation of inflammatory responses at the airway mucosa seen in asthma. However, overshooting immune reactions causes tissue damage that potentially impairs lung function and represents the basis of chronic inflammatory diseases, such as asthma, where pulmonary immunity is tightly regulated. AECs contribute to this regulation by releasing various anti-inflammatory cytokines. This, for example, holds true for TGF-β1, a cytokine of the TGF-β1-family, which is widely expressed by immune and stromal cells as well as by AECs [274,275,276]. TGF-β1 provides potent regulatory effects on a large variety of immune cells [281,282,283]. In particular, it suppresses the activity of TH1 as well as that of TH2 cells, macrophages, and NK cells and further promotes differentiation of FoxP3-expressing Tregs [284,285]. Allergen challenge leads to the activation of TGF-β signaling in both asthma patients and mice with experimental allergic asthma [283,286]. In line with its described ability to downregulate TH2 cell activity, adoptive transfer of OVA-specific and TGF-β1-overpressing T helper cells into mice with experimental, OVA-induced asthma leads to diminished allergic airway inflammation and AHR [287]. In turn, reduction of TGF-β1 expression in heterozygous tgfb1−/+ mice (since homozygous tgfb1−/− mice prematurely die from excessive inflammation [288]) results in increased TH2 type airway inflammation and aggravation of pathophysiologic hallmarks in mice sensitized and challenged with OVA [289]. However, since TGF-β1 does not only provide potent regulatory effects on immune cells but also acts on fibroblasts, myofibroblasts, and airway smooth muscle cells, it is suggested to play a pivotal role in those processes leading to airway remodeling and formation of AHR in asthma [290,291].

Another means by which the airway epithelium could participate in dampening inflammatory responses is the release of IL-37. Similar to its relatives of the IL-1 cytokine family IL-1 and IL-33, its biological activity is regulated by proteolytical cleavage. Expression of this cytokine has been reported for a variety of immune cells, especially monocytes, lymphocytes, and NK cells, and quite recently also for airway epithelial cells [292,293]. Its heterodimeric receptor is constituted by IL-18Rα and Tir-8 (formerly termed SIGIRR), which is expressed on most immune cells and many structural cells of the lung, including AECs [294,295,296]. Though expression of IL-37 has not been demonstrated for mice yet, murine IL-18Rα and Tir-8 are addressable by human recombinant IL-37 and provide anti-inflammatory effects comparable to those observed in human cells [294,295]. An impaired production of IL-37 by re-stimulated T cells and reduced serum levels of IL-37 from children with asthma have been reported [295,297]. The IL-37 level negatively correlates with IgE serum levels [298]. This is in line with the finding of reduced IL-37 expression and release by nasal epithelial cells from adult asthma patients who discontinued corticosteroid therapy [296]. IL-37 has been demonstrated to downregulate the proinflammatory activity of various innate immune cells [294], and it also inhibits allergic inflammation, as has been demonstrated by the local application of recombinant human IL-37 to mice suffering from experimental allergic asthma [295,299,300]. These effects could be attributed to an interaction/inhibition of IL-37 with NFκB and ERK1/2 signaling as induced by TSLP [299,300]. It also inhibits IL-4/IL-13-supported production of CCL11 and, therefore, downregulates eosinophil recruitment to the airways [301]. Very recently, it was further demonstrated in vivo and in vitro that IL-37 also counteracts the proinflammatory effects of both IL-1 and IL-33 on TH2 cell activation and on AECs [296]. This opens the possibility that AECs are able to regulate their own activation by IL-37 in an auto- or paracrine manner.

Briefly, AECs represent non-hematopoietic cells that carry out a broad spectrum of immune functions and, therefore, play a crucial role in orchestrating local immune responses in the lung. As such, they can be described as functionally related to the cells that constitute the innate immune system. Consequently, the question arises whether AECs are also capable of creating some kind of memory of pathogen contact, tissue damage, or inflammation, as it has been demonstrated for innate immune cells and termed trained immunity. Such an innate immune memory is based on epigenetic changes, such as DNA methylation, histone methylation, and acetylation [302], and it may contribute to an enhanced immunological fitness, for example, by enhanced expression of PRRs in order to sense danger signals, of cytokines and chemokines to regulate inflammation, as well as of antimicrobial peptides to directly target invading pathogens, and of cytokine receptors to increase the sensitivity to those signals. Given that, primary exposure to danger signals may somehow “prepare” AECs to face subsequent challenges by putting them “at attention” and enabling them to accelerate reactions, as has been proposed for children who are at low risk for severe COVID-19 [303]. On the other hand, innate immune memory could help to prevent overshooting cytokine production and, thus, inflammatory reactions. This is of particular interest since the average half-life of airway epithelial basal cells is described to exceed six months, predisposing them to become highly probable target cells for an innate immune memory [304]. Furthermore, AECs represent the most abundant cell type of the respiratory tract covering more than 90 m^2^ surface in humans, and together with the different macrophage populations (e.g., alveolar macrophages, interstitial or airway-associated macrophages), they are the most frequent cell type carrying our innate immune functions, such as pathogen recognition and cytokine release in the lung [305,306,307].

## 7. Trained Immunity of the Airway Epithelium

The first hints towards a training effect of non-immune cells were found in skin epithelial stem cells. These cells “remembered” exposure to the TLR7 ligand imiquimod by maintaining the chromosomal accessibility of both epidermal and inflammation genes for 180 days, which led to accelerated transcription in response to a repeated stimulus [308]. A little later, comparable effects were identified in nasal epithelia cells from patients suffering from chronic rhinosinusitis, an allergic disease of the upper respiratory tract that is characterized by allergic inflammation, remodeling of the epithelium, and abnormal tissue outgrowth, called polyps. Here, basal cells from the polyps displayed a memory of an allergic inflammatory milieu. This memory manifests by an enhanced sensitivity towards the TH2-type cytokines IL-4 or IL-13 and a more stem cell-like character of the cells in comparison to cells derived from non-polyp epithelia. The phenomenon was associated with accessible chromatin changes [309]. Remarkably, these cells displayed upregulation of the transcription factor ATF3, which is also observed during viral infection as well as in house dust mite allergy [310]. In a subsequent proof-of-concept study, Bigot et al. demonstrated that in vitro pre-exposure to flagellin of Pseudomonas aeruginosa alters the inflammatory response (e.g., the release of IL-6 or IL-8) of human bronchial epithelial cells to a second, non-related stimulus; in this case Aspergillus fumigatus conidia, Escherichia coli LPS, or *Stenotrophomonas maltophilia*. Again, the effects could be traced down to epigenetic modifications, such as histone acetylation and methylation, and this reprogramming may lead to both reduction or exacerbation of the inflammatory response, depending on the nature of the second stimulus [311]. These studies clearly indicate that non-immune cells, such as airway epithelial cells, are also able to install an innate immune memory. Remarkably, barrier and junctional defects, as well as ciliated cell hypoplasia, could also be observed in primary airway epithelial cells from smokers and COPD patients in air-liquid-interface cell culture, indicating that the memory of these cells could also include genes beyond those implemented in inflammation [312]. This could have a critical impact on the pathogenesis of chronic diseases of the airways since basal cells are not only involved in sensing and defense against pathogens but also play a significant role in the maintenance and renewal of the epithelial barrier. This is particularly true for cases of tissue damage and, thus, barrier impairment, when basal cells initiate a proinflammatory response as well as tissue repair by interaction with structural cells, such as fibroblasts as well as immune cells including macrophages, innate lymphoid tissue cells, and T cells. Thus, basal cells do not only release a broad range of growth factors, such as amphiregulin, epiregulin, neuregulin, and TGF-β, but also matrix metalloproteases (e.g., MMP-3, -9, and -11), that are central the injury-initiated epithelial to mesenchymal transition (EMT) [313,314]. The release of these MMPs leads to degradation and subsequent modification of the ECM that facilitates migration of epithelial progenitor cells towards the denuded epithelium. Once in place, these cells proliferate and differentiate in order to form a pseudostratified epithelium covering the damaged site—a process that is largely regulated by amphiregulin (AREG), in part by the basal cells in an autocrine fashion, and by insulin growth factor 1 (IGF-1), and IL-13 secreted [315,316,317]. If it turns out that basal cells also have the capacity to learn from previous encounters in order to respond faster and more robustly to barrier defects also by points of barrier repair, the role of these relatively long-lived cells in the processes leading to airway remodeling will gain even more importance.

## 8. Conclusions

In conclusion, the airway mucosa is the first compartment of the lung that encounters inhaled allergens, pathogens, and other environmental stressors that are potentially able to impair the prime function of the airway mucosa: to provide a barrier that separates the body from its environment in order to keep inner homeostasis. Consequently, the airway epithelium offers special structures to withstand these harmful factors. Constituted of multiple, densely interconnected cell types, several of which are secretory, the airway epithelium forms a sealed barrier that produces its own protective shield in the form of tight junctions, the glycocalyx/PCL, and an unstirred, discontinuous layer of mucous. Though this complex three-tiered structure is rather impressive, the airway epithelium does more than provide a passive barrier function. In particular, when this structure is challenged by invading pathogens and other noxae, it becomes clear that the airway epithelium plays a highly active role in the formation, strategic orientation, and control of the subsequent defense reaction. After pathogen/allergen contact, it does not only release active defense molecules, such as IgA and antimicrobial peptides, but calls the immune system for help and shares information on the type of invader by releasing a broad spectrum of proinflammatory cytokines. Once the pathogen/allergen has been cleared, the airway epithelium is also involved in the processes that initiate the resolution of the immune response and the reconstitution of destroyed tissues. Not long ago, all these functions were mostly attributed to cells of the innate or adaptive immune system. Now, it becomes increasingly obvious that the airway epithelium is a highly active part of the innate immune system and, as such, seems also to be “trained”, thereby possessing the ability to remember preceding contact with stressors. In light of this, it comes as no surprise that the airway epithelium and its ability to initiate, amplify, direct, and resolve inflammatory reactions is of vital importance to chronic inflammatory diseases, such as asthma.

## Figures and Tables

**Figure 1 cells-12-02208-f001:**
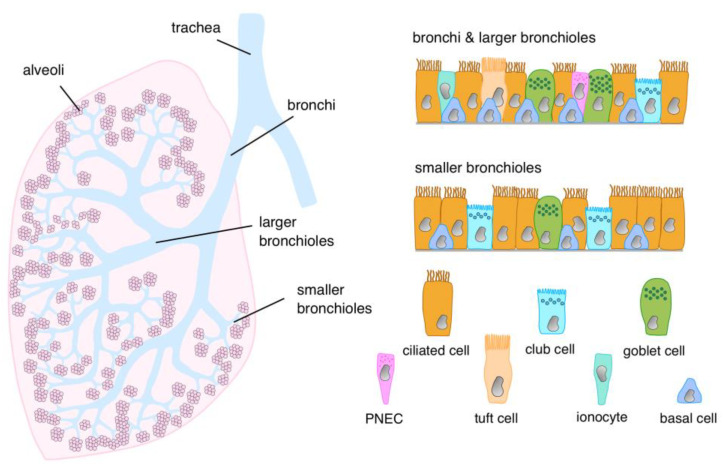
Epithelial cell types in different regions of the lower respiratory tract. The composition of the epithelial layer varies along the proximal-distal axis of the airways. While “classical” characterization of the epithelial lineage via histochemical methods has led to the identification of the more prominent cell types (basal cells, ciliated cells, goblet cells, club cells), novel approaches using single-cell RNA sequencing have ascertained the presence of less frequent cell types, such as tuft cells, neuroendocrine cells, and ionocytes.

**Figure 2 cells-12-02208-f002:**
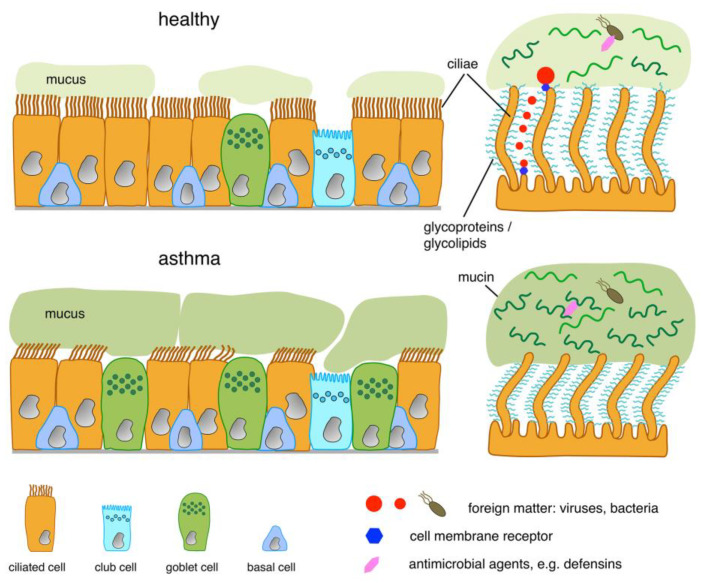
Structure of the luminal epithelial barrier in healthy and asthmatic airways. In the healthy state, microbes entrapped in mucus are attacked by antimicrobial agents (exemplified by a pink symbol). The viscous mucus gel is constantly moved onwards by the coordinated beating of the cilia. The cilia are covered by a dense meshwork of glycoproteins—the glycocalyx. Only particulate matter (e.g., viruses) (red), which are smaller in diameter than the glycocalyx pore size, can reach target receptors (blue) on the apical membrane; larger particles/viruses can only bind to receptors on cilial tips, respectively, in the outer part of the glycocalyx. In the asthmatic state, the composition of mucus is altered. Overproduced mucin MUC5AC inhibits antimicrobial activities, and higher mucin content leads to mucus thickening, dehydration, and shrinking of the periciliary layer. As a consequence, cilia are bent, and their beating becomes disordered. Movement of the thickened mucus plugs is impeded, resulting in obstruction of airways.

**Figure 3 cells-12-02208-f003:**
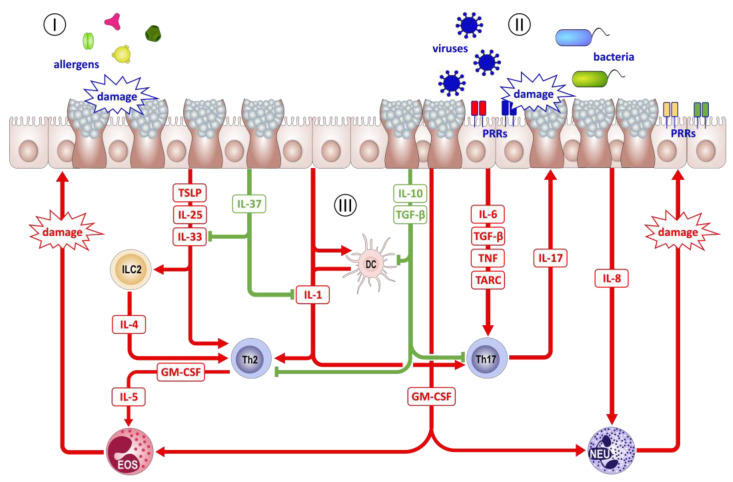
Immune functions of the airway epithelium in asthma pathogenesis. (I) Airway epithelial cells (AECs) react to damage by allergens with the release of mediators, such as thymic-stromal lymphopoietin (TSLP), interleukin (IL) 25, and IL-33, three factors that on the one hand stimulate innate lymphoid tissue cells (ILC2) to release IL-4, which in turn favors T helper 2 (TH2) cell differentiation, and on the other hand support production of TH2 type cytokines, such as IL-5, by TH2 cells. IL-5, together with GM-CSF, directs the recruitment of eosinophils to the inflammation site, which, in the case of allergic asthma, leads to further damage to the epithelium. (II) Contact with or infection with bacteria or respiratory viruses also leads to damage to AECs. AECs are able to recognize such pathogens through pattern recognition receptors (PRRs) and respond with the release of cytokines, such as IL-1, IL-6, and tumor necrosis factor (TNF) and the thymus and activation chemokine (TARC), which generally promote a local inflammatory reaction as well as the differentiation and recruitment of TH17 cells. TH17 cells, in turn, release proinflammatory IL-17, which drives the release of IL-8 by AECs, leading to chemotaxis of neutrophils and, thus, further damage to the airway mucosa. (III) While such a reaction to inhaled pathogens typically leads to aggravation of airway inflammation and exacerbation of established asthma, AECs are also capable of regulating inflammatory reactions. The release of IL-37 directly inhibits the proinflammatory effect of IL-1 as well as the TH2-favoring effects of IL-33, while the release of IL-10 and transforming growth factor (TGF) β generally downregulates the activity of all types of inflammation in order to foster repair processes. Red arrows indicate proinflammatory effects, whereas green lines indicate anti-inflammatory effects.

## Data Availability

Not applicable.

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
