# Peer review of "The Dual Role of the Airway Epithelium in Asthma: Active Barrier and Regulator of Inflammation"

_cells, 2023, doi:10.3390/cells12182208_

Round 1

Reviewer 1 Report (New Reviewer)

The article is dedicated to the airway epithelium in asthma, its active barrier functions and regulation of inflammation. The topic discussed is very interesting and crucial for the development of  treatment and prevention methods of asthma.

I would like to make a few comments:

1)    Lines 13-14: The phrase “In order to fulfill this task” perhaps, should be deleted.

2)    Lines 45-46: “Current concepts of asthma pathogenesis propose a strong involvement of allergens in disease initiation.”

Comment:  It's not quite right. Pollutants, tobacco smoking, cold temperature, respiratory infections, exposure to chemicals, and psychological factors, such as fear and anger, etc. may be asthma inducers:

 Lam HC, Li AM, Chan EY, et al. The short-term association between asthma hospitalisations, ambient temperature, other meteorological factors and air pollutants in Hong Kong: a time-series study. Thorax 2016;71:1097-1109.

World Health Organization. Asthma Fact sheet No 307. 2013. http://www.who.int/mediacentre/factsheets/fs307/en/

Thai, P.K., Zheng, Q., Phung, D. et al. The use of asthma and allergy medicines is associated with exposure to smoking. Nat Water 1, 443–450 (2023). https://doi.org/10.1038/s44221-023-00076-7

3)    Lines 53-57: “Once established, allergic spectrum disorders usually bias T helper cell differentiation towards the TH2 phenotype….”

Comment:  Several types of non-Th2 asthma have been described to date. Read and cite please:

Hudey SN, Ledford DK, Cardet JC. Mechanisms of non-type 2 asthma. Curr Opin Immunol. 2020 Oct;66:123-128. doi: 10.1016/j.coi.2020.10.002.

icciardolo FLM, Carriero V, Bertolini F. Which Therapy for Non-Type(T)2/T2-Low Asthma. J Pers Med. 2021 Dec 23;12(1):10. doi: 10.3390/jpm12010010.

4)        Line 96: “and other imprinting events”…  What does it mean? What imprinting events are discussed? The phrase “and other imprinting events” doesn't seem right.

5)        In table 1, the bibliography column should be formatted the same for all rows.

6)        Lines 261-281. It should be mentioned that the proteases contained in pollen can also destroy the tight junctions, breaking the integrity of the barrier:

Gaspar R, de Matos MR, Cortes L, Nunes-Correia I, Todo-Bom A, Pires E, Veríssimo P. Pollen Proteases Play Multiple Roles in Allergic Disorders. Int J Mol Sci. 2020 May 19;21(10):3578. doi: 10.3390/ijms21103578.

7)        When you are writing about TLRs, NLRs, and CLRs, please discuss the complex nature of innate immune receptors regulation of allergic inflammation. In particular, prolonged stimulation with low doses of innate immunity receptors agonists before sensitization by an allergen reduces the severity of the allergic process. In the case of innate immunity receptors stimulation together with the action of an allergen, allergic inflammation increases in asthma model.

Please discuss and cite the article:

Guryanova, S.V.; Gigani, O.B.; Gudima, G.O.; Kataeva, A.M.; Kolesnikova, N.V. Dual Effect of Low Molecular Weight Bioregulators of Bacterial Origin in Experimental Model of Asthma. Life 2022, 12, 192. https://doi.org/10.3390/life12020192

8)        Lines 557-582. Cases with Th1 and Th17 asthma subtypes have been described. This subtypes are steroid-resistant.

9)        Line 629: ”modulated: While..”

Comment:  “while” should be written with small letter.

10)     Lines 652-653: “This offers the possibility of a vicious-cycle-type feedback loop fostering a TH2-type local micro-milieu [273].”

Comment: Pease add the word “positive”: “This offers the possibility of a vicious-cycle-type positive feedback loop fostering a TH2-type local micro-milieu [273].

11)     Discussion: “Not long ago, all these functions have mostly been attributed to cells of the innate or adaptive immune system. Now it becomes more and  more obvious that the airway epithelium is a highly active part of the innate immune system”.

Comment: It is generally accepted that all mucosal surfaces belong to the innate immune system:

Nochi T, Kiyono H. Innate immunity in the mucosal immune system. Curr Pharm Des. 2006;12(32):4203-13. doi: 10.2174/138161206778743457.

Radicioni, G., Cao, R., Carpenter, J. et al. The innate immune properties of airway mucosal surfaces are regulated by dynamic interactions between mucins and interacting proteins: the mucin interactome. Mucosal Immunol 9, 1442–1454 (2016). https://doi.org/10.1038/mi.2016.27

Author Response

Dear Professor Pilette, dear Reviewer 1,

We are glad that our manuscript is still of interest of your editorial board and we would like to thank you for giving us a third opportunity to revise our review article. We further would like to express our appreciation for all the work, you and especially the reviewer have put in our manuscript. So, please, find attached the revised version our review manuscript entitled “The dual role of the airway epithelium in asthma: active barrier and regulator of inflammation”, which we would like to re-submit to your special issue “Mucosal Immunity in Respiratory Diseases” of the renowned Cells.

We now hope that it is finally suitable to the readership of the special issue “Mucosal Immunity in Respiratory Diseases” of the renowned Cells.  Therefore, we would be very grateful for publishing our manuscript.

Please, find in the following our point-by-point reply to the comments of the reviewer’s.

On behalf of all authors and sincerely yours,

Michael Wegmann, PhD

point-by-point reply to reviewer 1

The article is dedicated to the airway epithelium in asthma, its active barrier functions and regulation of inflammation. The topic discussed is very interesting and crucial for the development of treatment and prevention methods of asthma.

I would like to make a few comments:

Comment: 1)    Lines 13-14: The phrase “In order to fulfill this task” perhaps, should be deleted.

Reply: The respective phrase has been changed according to the suggestion of the reviewer.

Comment: 2)    Lines 45-46: “Current concepts of asthma pathogenesis propose a strong involvement of allergens in disease initiation.” It's not quite right. Pollutants, tobacco smoking, cold temperature, respiratory infections, exposure to chemicals, and psychological factors, such as fear and anger, etc. may be asthma inducers:

Lam HC, Li AM, Chan EY, et al. The short-term association between asthma hospitalisations, ambient temperature, other meteorological factors and air pollutants in Hong Kong: a time-series study. Thorax 2016;71:1097-1109.

World Health Organization. Asthma Fact sheet No 307. 2013. http://www.who.int/mediacentre/factsheets/fs307/en/

Thai, P.K., Zheng, Q., Phung, D. et al. The use of asthma and allergy medicines is associated with exposure to smoking. Nat Water 1, 443–450 (2023). https://doi.org/10.1038/s44221-023-00076-7 

Reply: Although there is nothing wrong with our statement that current concepts propose allergens to be strongly involved in the pathogenesis of asthma, our formulation could have possibly been more precisely. One has to distinguish between those factors that are involved in the initiation of the disease (or by other means the sensitization to allergens, the following allergic inflammation of the airways, and the subsequent pathophysiologic consequences) and those factors that have been associated with the acute worsening of the already established disease (which means inducers of acute asthma attacks or acute exacerbation). While the first factors are described as risk factors and foremost involve allergens (e.g. with inherent protease activity etc.), respiratory infection, and of course a genetic predisposition, the factors mentioned by the reviewer (pollutants, tobacco smoking, cold temperature, respiratory infections, exposure to chemicals, and psychological factors, such as fear and anger) are commonly associated with acute worsening of asthma. We are absolutely aware that mouse models have been published that e.g. ozone, toluene, or LPS as adjuvants to sensitize animals. However, these studies still await confirmation by clinical studies. We implemented this difference into the revised passage.

Comment: 3)    Lines 53-57: “Once established, allergic spectrum disorders usually bias T helper cell differentiation towards the TH2 phenotype….” Several types of non-Th2 asthma have been described to date. Read and cite please:

Hudey SN, Ledford DK, Cardet JC. Mechanisms of non-type 2 asthma. Curr Opin Immunol. 2020 Oct;66:123-128. doi: 10.1016/j.coi.2020.10.002. 

Ricciardolo FLM, Carriero V, Bertolini F. Which Therapy for Non-Type(T)2/T2-Low Asthma. J Pers Med. 2021 Dec 23;12(1):10. doi: 10.3390/jpm12010010.

Reply: We have specified the TH2 bias as a feature of allergic asthma, which, to our knowledge, remains state-of-the art understanding. However, as the reviewer rightfully states, other than allergic (or TH2-high) asthma phenotypes exist. We have therefore included a short paragraph on these cases and have added references suggested by the reviewer.

Comment: 4)        Line 96: “and other imprinting events”…  What does it mean? What imprinting events are discussed? The phrase “and other imprinting events” doesn't seem right.

Reply: We agree with the reviewer that this phrasing is imprecise and therefore have deleted this part of the sentence.

Comment: 5)        In table 1, the bibliography column should be formatted the same for all rows.

Reply: Response: Indeed a formatting error appeared during the revision process of the manuscript. The bibliography column in table 1 was corrected according to the last rows of this table. We thank the reviewer for this helpful comment.

Comment: 6)        Lines 261-281. It should be mentioned that the proteases contained in pollen can also destroy the tight junctions, breaking the integrity of the barrier:

Gaspar R, de Matos MR, Cortes L, Nunes-Correia I, Todo-Bom A, Pires E, Veríssimo P. Pollen Proteases Play Multiple Roles in Allergic Disorders. Int J Mol Sci. 2020 May 19;21(10):3578. doi: 10.3390/ijms21103578.

Reply: We thank the reviewer for pointing out this important fact which we had indeed failed to mention. We have now included this finding with appropriate original references.

Comment: 7)        When you are writing about TLRs, NLRs, and CLRs, please discuss the complex nature of innate immune receptors regulation of allergic inflammation. In particular, prolonged stimulation with low doses of innate immunity receptors agonists before sensitization by an allergen reduces the severity of the allergic process. In the case of innate immunity receptors stimulation together with the action of an allergen, allergic inflammation increases in asthma model. Please discuss and cite the article:

Guryanova, S.V.; Gigani, O.B.; Gudima, G.O.; Kataeva, A.M.; Kolesnikova, N.V. Dual Effect of Low Molecular Weight Bioregulators of Bacterial Origin in Experimental Model of Asthma. Life 2022, 12, 192. https://doi.org/10.3390/life12020192

Reply: We thank the reviewer for this helpful comment. We added a short discussion concerning the multidirectional effects of PRR signaling on the development of allergic diseases and included the recommended manuscript by Guryanova et al.

Comment: 8)        Lines 557-582. Cases with Th1 and Th17 asthma subtypes have been described. This subtypes are steroid-resistant.

Reply: We thank the reviewer for this helpful comment. We added information on TH1 and TH17 dominated asthma subtypes into the respective paragraph of the revised manuscript.

Comment: 9)        Line 629: ”modulated: While..”   “while” should be written with small letter.

Reply: We have changed the phrase as suggested by reviewer.

Comment: 10)     Lines 652-653: “This offers the possibility of a vicious-cycle-type feedback loop fostering a TH2-type local micro-milieu [273].” Pease add the word “positive”: “This offers the possibility of a vicious-cycle-type positive feedback loop fostering a TH2-type local micro-milieu [273].

Reply: We believe the addition of the word “positive” in this context to be misleading and would rather avoid it.

Comment: 11)     Discussion: “Not long ago, all these functions have mostly been attributed to cells of the innate or adaptive immune system. Now it becomes more and  more obvious that the airway epithelium is a highly active part of the innate immune system”.

Reply: It is generally accepted that all mucosal surfaces belong to the innate immune system:

Nochi T, Kiyono H. Innate immunity in the mucosal immune system. Curr Pharm Des. 2006;12(32):4203-13. doi: 10.2174/138161206778743457.

Radicioni, G., Cao, R., Carpenter, J. et al. The innate immune properties of airway mucosal surfaces are regulated by dynamic interactions between mucins and interacting proteins: the mucin interactome. Mucosal Immunol 9, 1442–1454 (2016). https://doi.org/10.1038/mi.2016.27

Reply: We are absolutely in line with the reviewer and really appreciate this view of the airway mucosa. However, over the last decade we all remember more than lively and controversial discussions on that issue with very harsh statements that did not include epithelial cells to the immune system. Since even standard text books as well as large reviews on the immune system still describe epithelial surfaces as a “first line of defense” that above all functions as a physical barrier preventing pathogens from entering the organism, we chose the given formulation that avoids distinct allocation of AECs but rather focusses on its functions. In line, we spent large parts of the review to collect and describe these functions and to emphasize the contribution of AECs to those complex networks that constitute our immune system. Last but not least especially the attribution of trained immunity to AECs is very new and includes just the first few studies on it that have been published in the last 1-2 years. Thus, especially this function is not common sense, so that we really would like to stay with the original formulation.

Reviewer 2 Report (New Reviewer)

The aim of the paper written by Andreas Frey et al. is to construct the dual role of the airway epithelium in asthma: active barrier and regulator of inflammation. To reach these aims, a narrative review was conducted. The authors want to outline how the airway epithelial layer is equipped to deal with this awkward position and what the short- and long-term effects may be, for the epithelium itself, the neighboring stroma and lumen and their inhabitants and lastly, for the entire lung and its afferent bronchi. They concluded that t the airway epithelium and its ability to initiate, amplify, direct, and resolve inflammatory reactions is of central importance to chronic inflammatory diseases such as asthma. Following are concerns and several suggestions given to improve the report:

1.          In Figure 3, the regulation of immune function in respiratory epithelial cells is very clear, but in line 516 of the text, it is mentioned that viruses and bacteria produce a series of cytokines such as IL-6, TNF, etc. through the PRR pathway, but in this series IL-1 is not marked (IL-1 appears on the allergens pathway: on the left of Figure 3).

2.          In the line 572 of this article, it is mentioned that "The main source of IL-17 appears to be TH17 cells but not the airway epithelium----" and in the line 612, it is mentioned that "IL-25, a member of the IL-17 cytokine family , is produced by AECs as well as by TH2 cells----" seems to be somewhat contradictory?

3.          Do the Type 1 interferons mentioned in line 540 of this article contain IFN-r?.

4.          Lines 546 to 548 of this article mention: the mediator secreted by tracheal epithelial cells has an impact on asthma, will it be related to the severity of asthma? Also, is it possible to attach references to this paragraph?

5.          On pages 14 to 15, the relationship between PSLP, IL25 and IL33 and the airway epithelial cells is very clearly described. Can you make a simple diagram according to the type of Figure 3?

6.          What does "one the one hand" mean in line 581 of this article "one the one hand they play a significant role……"?

7.          In line 526 "Expressing a wide spectrum of PRRs (e.g., TLRs, NLRs, RAGE) airway AECs are equipped -----" whether to change "---airway AECs are equipped---" to "---AECs are equipped---"?

Quality of English Language is fine

Author Response

Dear Professor Pilette, dear Reviewer 2,

We are glad that our manuscript is still of interest of your editorial board and we would like to thank you for giving us a third opportunity to revise our review article. We further would like to express our appreciation for all the work, you and especially the reviewer have put in our manuscript. So, please, find attached the revised version our review manuscript entitled “The dual role of the airway epithelium in asthma: active barrier and regulator of inflammation”, which we would like to re-submit to your special issue “Mucosal Immunity in Respiratory Diseases” of the renowned Cells.

We now hope that it is finally suitable to the readership of the special issue “Mucosal Immunity in Respiratory Diseases” of the renowned Cells.  Therefore, we would be very grateful for publishing our manuscript.

Please, find in the following our point-by-point reply to the comments of the reviewer’s.

On behalf of all authors and sincerely yours,

Michael Wegmann, PhD

point-by-point reply

The aim of the paper written by Andreas Frey et al. is to construct the dual role of the airway epithelium in asthma: active barrier and regulator of inflammation. To reach these aims, a narrative review was conducted. The authors want to outline how the airway epithelial layer is equipped to deal with this awkward position and what the short- and long-term effects may be, for the epithelium itself, the neighboring stroma and lumen and their inhabitants and lastly, for the entire lung and its afferent bronchi. They concluded that t the airway epithelium and its ability to initiate, amplify, direct, and resolve inflammatory reactions is of central importance to chronic inflammatory diseases such as asthma. Following are concerns and several suggestions given to improve the report:

Comment: 1.          In Figure 3, the regulation of immune function in respiratory epithelial cells is very clear, but in line 516 of the text, it is mentioned that viruses and bacteria produce a series of cytokines such as IL-6, TNF, etc. through the PRR pathway, but in this series IL-1 is not marked (IL-1 appears on the allergens pathway: on the left of Figure 3).

Reply: We thank the reviewer for this helpful comment. Indeed, the effects of (epithelial derived) are not restricted to promotion of TH2-type immune responses but also affect all other types of inflammation e.g. by promoting TH17 differentiation and activation and consequently neutrophilic inflammation. We have revised figure 3 accordingly.

Comment: 2.          In the line 572 of this article, it is mentioned that "The main source of IL-17 appears to be TH17 cells but not the airway epithelium----" and in the line 612, it is mentioned that "IL-25, a member of the IL-17 cytokine family , is produced by AECs as well as by TH2 cells----" seems to be somewhat contradictory?

Reply: Indeed, the literature describes TH17 cells and not AECs to be the major source of IL-17, whereas AECs and TH2 cells have been shown to be potent producers of IL-25. This is not contradictory since IL-17, also known as IL-17A, and IL-25, also known as IL-17E, actually represent two different cytokines belonging to the same cytokine family. Perhaps, we have used a somehow unclear formulation by including the phrase that IL-25 is a member of the IL-17 cytokine family. We therefore included the alternative name IL-17E to the manuscript and labelled IL-17 as IL-17A throughout the text in order to underline the different character of both cytokines.

Comment: 3.          Do the Type 1 interferons mentioned in line 540 of this article contain IFN-r?.

Reply: Type I interferons include IFNs -a, -b, -e, -k, -l, and -w, of those we mentioned only 3 (-a, -b, and -l) that have been implemented in the pathogenesis of asthma. However, an IFN-r is not known to us. Despite extensive search of the available literature we were not able to find IFN-r or articles describing its implementation in anti-viral or allergic immune responses. Nevertheless, we corrected the classification of IFN-l from type I to type III interferons and hope to have addressed this point appropriately.

Comment: 4.          Lines 546 to 548 of this article mention: the mediator secreted by tracheal epithelial cells has an impact on asthma, will it be related to the severity of asthma? Also, is it possible to attach references to this paragraph? 

Reply: Indeed, we did not confer to the mediator release of the trachea exclusively, but described the mediator release by AECs in general. In the respective lines we gave few cytokines and interferons as examples. In the following appr. 180 lines we describe the effect of several – but of course not all – mediators released by the AECs in response to pathogen contact and its respective effects on the pathogenesis of asthma. By doing this, we concentrated mainly on the effects of cytokines in order to describe the regulatory functions of AECs, which previously have been mainly attributed to immune cells and especially T cells. Concerning the effect of type I and III interferons, these mediators have also an effect on asthma pathogenesis, which is perhaps due to their autocrine and paracrine function different from that of cytokines. Since the manyfold revisions of our article led to its considerable length with more than 300 references and many figures, we decided to omit a new section on the effects of interferons and instead referred to a comprehensive and well written review on this subject. We hope that this will find the approval of the reviewer.

Comment: 5.          On pages 14 to 15, the relationship between PSLP, IL25 and IL33 and the airway epithelial cells is very clearly described. Can you make a simple diagram according to the type of Figure 3?

Reply: Maybe the reviewer missed it, but the effects of TSLP, IL-25, and IL-33 have been implemented in figure 3 like all other key cytokines (e.g. IL-1, IL-4, IL-5, IL-6, IL-37, TNF, etc.) involved in the regulation of the inflammatory processes underlying asthma development. Of course this figure does not include all effects of the mentioned mediators on all cells participating in asthma pathogenesis, but the key effects of the mentioned cytokines on key cells are already included. We imply that this grade of didactic reduction helps the reader in a comprehensive way to register the fundamental functionality of the airway epithelium in the inflammatory processes of asthma and hope that this will the approval of the reviewer.

Comment: 6.          What does "one the one hand" mean in line 581 of this article "one the one hand they play a significant role……"?

Reply: The sentence has been rephrased for clarity.

Comment: 7.          In line 526 "Expressing a wide spectrum of PRRs (e.g., TLRs, NLRs, RAGE) airway AECs are equipped -----" whether to change "---airway AECs are equipped---" to "---AECs are equipped---"?

Reply: We thank the reviewer for this helpful comment. The respective sentence has been revised accordingly.

Reviewer 3 Report (New Reviewer)

It is a well written review article on AECs and their role in health and asthmatic disease. Here are my comments:

-Please add cells of trachea also in Table. 1

-Please label different structures in Fig. 2.

-Please label different receptors present on surface of AECs, e.g. PAR2, TLRs relevant to asthma 

Ok

Author Response

Dear Professor Pilette, dear reviewer 3,

We are glad that our manuscript is still of interest of your editorial board and we would like to thank you for giving us a third opportunity to revise our review article. We further would like to express our appreciation for all the work, you and especially the reviewer have put in our manuscript. So, please, find attached the revised version our review manuscript entitled “The dual role of the airway epithelium in asthma: active barrier and regulator of inflammation”, which we would like to re-submit to your special issue “Mucosal Immunity in Respiratory Diseases” of the renowned Cells.

We now hope that it is finally suitable to the readership of the special issue “Mucosal Immunity in Respiratory Diseases” of the renowned Cells.  Therefore, we would be very grateful for publishing our manuscript.

Please, find in the following our point-by-point reply to the comments of the reviewer’s.

On behalf of all authors and sincerely yours,

Michael Wegmann, PhD

It is a well written review article on AECs and their role in health and asthmatic disease. Here are my comments:

Comment: -Please add cells of trachea also in Table. 1

Reply: Of course airway epithelial cells cover the surface of the trachea as well as in the attached bronchi and bronchiole, but also of the nasopharynx. Since our review focusses on the role of the airway epithelium in asthma pathogenesis, a disease that is characterized by symptoms arising in the airways below but not in the trachea, e.g. the asthma attack, we decided not to include the trachea in order to keep focused on the centrally involved sections of the breathing system and hope that this decision will find the approval of the reviewer.

Comment: -Please label different structures in Fig. 2.

Reply: We thank the reviewer for this helpful comment. The labeling has been altered according to his suggestion and gained considerable readability.

Comment: -Please label different receptors present on surface of AECs, e.g. PAR2, TLRs relevant to asthma 

Reply: We are totally in line with the reviewer. Therefore, we labelled the PRRs on the surface of AECs in figure 3, which depicts their (regulatory) immune functions. We think that here the labelling is most appropriate. Furthermore, we decided to use to summarize all the respective receptors as PRRs and to leave out the detailed list of all. Since only sparse data on the expression of PAR2 by airway epithelial cells in humans is available yet, we further decided not to include this receptor yet.

This manuscript is a resubmission of an earlier submission. The following is a list of the peer review reports and author responses from that submission.

Round 1

Reviewer 1 Report

This review covers the dual role of the airway epithelium in the immune response to insults and as a physical barrier, focusing on epithelial IgA and the regulation of DAMP/PAMPs by the epithelium. It focuses on the role of tight junctions and the glycocalyx in barrier function, and how the epithelium releases and regulates DAMP/PAMPs and the contribution of the epithelium to soluble IgA-mediated B cell responses. While giving a great summary of the research and identifying some epithelial dysfunctions, the link between these dysfunctions and the clinical manifestations of asthma is not clear. Overall, this review covers these two sections in depth, but is not suitable for publication in its current form. 

This review appears to duplicate a prior review published by the authors in Frontiers in Immunology (https://doi.org/10.3389/fimmu.2020.00761), and the contribution of the review to the field beyond the previous publication is not clear. While this review does include up-to-date information regarding barrier function and immune functions of the airway epithelium, these are not novel concepts and both of these topics are frequently the focus of reviews (including by the authors). 

Additionally, there are specific issues within this manuscript that the authors should address:

  1. There is a clear lack of referencing to support statements made throughout this review, but particularly in sections 1, 2, and 3.
  2. Addtionally, there are lots of references in this review to other review articles, rather than the primary source that demonstrates the concept or pathway being described.
  3. The scientific writing throughout needs to be improved, with the length of some sentences making it difficult to determine what the authors are trying to convey (e.g., lines 54-57, 176-181)
  4. Table 1 - there are a number of studies utilising cutting-edge single-cell RNA sequencing in the airway to clearly identify cell types along the bronchi to bronchiole and describe the cellular composition of the airway throughout the structures of the lung. The authors should include some quantification from these seminal studies (PMID: 33321047, PMID: 32726565, PMID: 33208946)
  5. The authors make the supposition that mucus obstruction may lead to airway obstruction (line 186) - defective mucociliary clearance has been observed in mild asthma since the 80s/90s, with case reports of death due to mucus plugging since the 1960s, and the role of mucus dysfunction has been comprehensively reviewed in asthma (PMID: 29186064).
  6. Line 188 - The section on the glycocalyx seems irrelevant to asthma and the airway, considering no differences in asthma have been reported, little has been done in the airway, and there have been a number of successful viral vector based gene therapies since the cited failed study, both in vitro and in vivo (PMID: 30538635)
  7. This review could greatly benefit from figures describing the pathways/responses to stimuli that have been described in the text
  8. There are a number of instances throughout the text where something is "assumed", "apparently", "so-called" - and it is not clear whether this is the authors opinion or the result of the study being cited.
  9. Section 4 seems to focus mostly on the effect of sIgA in immune cells, and the role of airway epithelial cells and effects on IgA should be expanded upon
  10. This review should touch on  the role of the airway epithelium in neutrophillic asthma, and how it impacts recruitment and inflammatory signalling

Author Response

Reply to 

Reviewer No. 1:

R: This review covers the dual role of the airway epithelium in the immune response to insults and as a physical barrier, focusing on epithelial IgA and the regulation of DAMP/PAMPs by the epithelium. It focuses on the role of tight junctions and the glycocalyx in barrier function, and how the epithelium releases and regulates DAMP/PAMPs and the contribution of the epithelium to soluble IgA-mediated B cell responses. While giving a great summary of the research and identifying some epithelial dysfunctions, the link between these dysfunctions and the clinical manifestations of asthma is not clear. Overall, this review covers these two sections in depth, but is not suitable for publication in its current form. 

R: This review appears to duplicate a prior review published by the authors in Frontiers in Immunology (https://doi.org/10.3389/fimmu.2020.00761), and the contribution of the review to the field beyond the previous publication is not clear. While this review does include up-to-date information regarding barrier function and immune functions of the airway epithelium, these are not novel concepts and both of these topics are frequently the focus of reviews (including by the authors). 

C: I greatly appreciate the time and work the reviewer has spent into appraising our manuscript and thank him for his valuable comments. I further welcome any critics, suggestions and comments that help us to improve our review and we are experienced enough to take even harshly formulated critics not personally. However, if I am confronted with the reproach of scientific misconduct because the reviewer accuses my co-authors and me to “duplicate” our own work, which implies plagiarism of our previous article on a comparable thematic issue. I am not going to tolerate this. I feel obliged to strictly point out that not a single phrase of the mentioned review article published in the Frontiers of Immunology has been duplicated, copied or reused by us in our present article.
As already mentioned before, both articles in part cover the same thematic issue, namely the immune functions of the airway epithelium and reach out to its implementation into asthma pathogenesis. It is common knowledge that a review article summarizes the current state of understanding on a topic, which further implies that the authors of such an article are at least familiar to that topic or ideally experts in this field. If this is mandatory and scientific authors are willing to compose more than one review in their field of expertise, their review articles of course display a certain degree of relation. Furthermore, it is in the nature of a review article that it does not provide novel concepts but a comprehensive overview of the actual state of knowledge about a special topic. Otherwise, there are other formats available (e.g. letter to the editor, editorials, etc.) to do as the reviewer suggests. Thus, the reviewer’s general criticism on the content novelty of our review article is quite not understandable to me. Having this clarified, I would like to proceed with addressing the specific critics reviewer no. 1 has raised.

Additionally, there are specific issues within this manuscript that the authors should address:

  1. There is a clear lack of referencing to support statements made throughout this review, but particularly in sections 1, 2, and 3.

C: We understand this point and agree with the reviewer. We therefore, added a considerable number of new references to sections 1,2, and 3.

  1. Addtionally, there are lots of references in this review to other review articles, rather than the primary source that demonstrates the concept or pathway being described.

C: Again, we agree with the reviewer (and also reviewer no. 2, who remarked the same), We have therefore replaced the majority of review article by original articles in the revised version.

  1. The scientific writing throughout needs to be improved, with the length of some sentences making it difficult to determine what the authors are trying to convey (e.g., lines 54-57, 176-181)

C: In order to improve legibility the whole text has been checked for difficult-to-understand phrases and multi-clause sentences. Wherever possible without compromising clarity the writing was streamlined. Lines 54-57 (of the original manuscript), which are in the first part of the introduction and consist of three sentences. Two of these sentences contain a relative clause. One sentence is a main clause. While the first two sentences are about 1,5 lines in length, the third one was longer. It is now split into two sentences. The same was done for the text in lines 176-181 (of the original manuscript).

  1. Table 1 - there are a number of studies utilising cutting-edge single-cell RNA sequencing in the airway to clearly identify cell types along the bronchi to bronchiole and describe the cellular composition of the airway throughout the structures of the lung. The authors should include some quantification from these seminal studies (PMID: 33321047, PMID: 32726565, PMID: 33208946)

C: The authors agree with the reviewer that the single-cell RNA sequencing technique has provided novel insight into the distinct cell populations and sub-populations of the human airways. However, while the data obtained in these studies give a plethora of valuable information about the cell types and their specific characteristics, quantitative information on relative abundancy of the different "basic" cell types is hard to be gained from them - mainly because of the complexity and heterogeneity of transcripts analyzed. We have nevertheless re-evaluated a large number of original single-cell RNA sequencing studies analyzing the human airway cell types - including the ones mentioned by the reviewer - and have added pertinent data and references in the text and table where applicable.

  1. The authors make the supposition that mucus obstruction may lead to airway obstruction (line 186) - defective mucociliary clearance has been observed in mild asthma since the 80s/90s, with case reports of death due to mucus plugging since the 1960s, and the role of mucus dysfunction has been comprehensively reviewed in asthma (PMID: 29186064).

C: We are well aware of the fact that defects in mucociliary clearance of asthmatics is known for decades already. Yet, it would have been a neglect to not address it in the context of the reviewed topic. As the finding can be viewed a commonplace, we felt it to be justified to cite a review, the one mentioned by the reviewer (PMID: 29186064).

  1. Line 188 - The section on the glycocalyx seems irrelevant to asthma and the airway, considering no differences in asthma have been reported, little has been done in the airway, and there have been a number of successful viral vector based gene therapies since the cited failed study, both in vitro and in vivo (PMID: 30538635)

C: Whether or not the airway epithelial cell glycocalyx is directly relevant or irrelevant to asthma (pathogenesis) is indeed difficult to address at this point in time. The study of Button et al. published in Science and cited in this review, demonstrates that the periciliary protein meshwork indeed may act as a barrier on the healthy epithelium whereas in case of the mucin-enriched mucus of asthmatics the periciliary protein meshwork (glycocalyx) may be impaired. We agree with the reviewer that viral vector based gene therapy approaches had been successful in recent years. Yet, it is still subject of debate whether the airway surface must be prepared in order to become more sensitive to viral transfection in order to achieve good gene therapy results. In particular, very small viruses whose receptor resides in the glycocalyx seem to perform very good. This supports the results of Button et al. that the periciliary protein meshwork (glycocalyx) should have a filtering effect.

  1. This review could greatly benefit from figures describing the pathways/responses to stimuli that have been described in the text

C: We thank the reviewer for this helpful comment. As suggested we have added two figures to our review article, one is illustrating the micro-architecture of the airway mucosa, the other depicts the regulatory network of the local immune response focusing on the role of the airway epithelium.

  1. There are a number of instances throughout the text where something is "assumed", "apparently", "so-called" - and it is not clear whether this is the authors opinion or the result of the study being cited.

C: We agree with the reviewer that ambigous wording should be avoided in scientific writing. Consequently, vague terms such as "assumed", "apparently", "so-called" and the like have been eliminated as far as possible. Where not possible phrasing now states clearly whether we or others claim something.

  1. Section 4 seems to focus mostly on the effect of sIgA in immune cells, and the role of airway epithelial cells and effects on IgA should be expanded upon

C: In terms of IgA-based mucosal immunity the main role of the epithelium is the translocation of basolaterally offered dimeric IgA onto the apical site. Except for neutralization and/or clearance of intracellular pathogens by IgA en route to the apical surface, the epithelium does not play any further roles for IgA. However, its ability and capacity to transport the dimeric IgA seems crucial for airway health. Thus, the actual contribution of the sIgA mediated mucosal immunity is not the IgA itself but rather the pIgR expressed by the epithelium and downregulated in case of asthma. Concerning apically released secretory IgA no explicit information on IgA function and the mechanism thereof in asthmatic airways is available in addition to the one provided in the manuscript.

  1. This review should touch on the role of the airway epithelium in neutrophillic asthma, and how it impacts recruitment and inflammatory signalling

C: This is indeed a very good point! According to the suggestion of the reviewer we have added a section on the involvement of the airway epithelium in the development of neutrophilic asthma and neutrophil recruitment to the site of inflammation, respectively.

Reviewer 2 Report

The authors provide a review on airway epithelial cells cells and their multitude of functions - physiological and pathological. Overall the review is detailed, but at portions very dense. It would overall benefit from a few more stylistic elements (i.e. a Figure, additional Tables). As this is a review, scientifically there are no comments. However - when reviewing a review, I personally focus always on the papers that where cited. My comments therefore largely focus on those. 

Positive things first - I think the review was extensive, and very detailed. The few minor comments are things that spontaneously came to mind when I was reading and looking at the referenced articles. The few minor comments I have should be easy to implement 

There are a few minor comments I have: 

  • many of the citations are review articles themselves, I feel the job of a review is to summarize the current literature, not give an overview of other review articles. 
  • in turn - this review citing let's original authors that contributed to the field fall short (i.e. Dagher/Pretolani 2020; Danahay/Jaffe 2015; Kuperman/Erle 2002). In particular these three papers have linked crucial mediators of asthma (IL13, IL33 etc.) directly to epithelial biology and maintenance. 
  • all citations on macrophages focus on alveolar macrophages. It has been shown that alveolar macrophages have distinct molecular signatures to airway macrophages (Gibbings/Jakubzick 2017; Engler/Rock 2020). 
  • while the architecture is explained in very great detail - a schematic picture might be more visual and easier to understand.
  • Also Citation (1) does not seem to fit the topic? Chandan, J.S. Improving Global Surveillance of Gender-Based Violence. Lancet 2020, 396, 1562, doi:10.1016/S0140- 648 6736(20)32319-9

Author Response

Reply to 

Reviewer No. 2:

R: The authors provide a review on airway epithelial cells cells and their multitude of functions - physiological and pathological. Overall the review is detailed, but at portions very dense. It would overall benefit from a few more stylistic elements (i.e. a Figure, additional Tables). As this is a review, scientifically there are no comments. However - when reviewing a review, I personally focus always on the papers that where cited. My comments therefore largely focus on those. 

Positive things first - I think the review was extensive, and very detailed. The few minor comments are things that spontaneously came to mind when I was reading and looking at the referenced articles. The few minor comments I have should be easy to implement 

There are a few minor comments I have: 

  • many of the citations are review articles themselves, I feel the job of a review is to summarize the current literature, not give an overview of other review articles. 

C: We absolutely agree with this view and, therefore, replaced the initially cited review articles whenever it was possible in the revised version of our review article.

  • in turn - this review citing let's original authors that contributed to the field fall short (i.e. Dagher/Pretolani 2020; Danahay/Jaffe 2015; Kuperman/Erle 2002). In particular these three papers have linked crucial mediators of asthma (IL13, IL33 etc.) directly to epithelial biology and maintenance. 

C: Accordingly, we have cited the original articles whenever possible in the revised version of our review article and additionally cited the three articles mentioned by the reviewer.

  • all citations on macrophages focus on alveolar macrophages. It has been shown that alveolar macrophages have distinct molecular signatures to airway macrophages (Gibbings/Jakubzick 2017; Engler/Rock 2020).

C: We thank the reviewer for raising this interesting point. Since it is our intention to focus on the immune functions of the airway epithelium and its involvement in the pathogenesis of bronchial asthma, we have reduced those parts that touch on the role of alveolar, interstitial or airway-associated macrophages. Macrophages are now only mentioned three times, mostly as cells targets cells of distinct cytokine signals. We hope that this will improve the focus and readability of our review article.

  • while the architecture is explained in very great detail - a schematic picture might be more visual and easier to understand.

C: We absolutely agree with the reviewer and, therefore, have added a figure on the micro-architecture of the airway mucosa to the revised version of our review article.

  • Also Citation (1) does not seem to fit the topic? Chandan, J.S. Improving Global Surveillance of Gender-Based Violence. Lancet 2020, 396, 1562, doi:10.1016/S0140- 648 6736(20)32319-9

C: We thank the reviewer for finding this “mis-citation”. Indeed, we made a mistake and fixed it by replacing the citation with the right one.

Round 2

Reviewer 1 Report

I have reviewed the authors responses to comments and changes they've made to this manuscript. While changes fixing issues highlighted were made by the authors, and I appreciate the time and energy that goes into writing a review manuscript, I cannot agree to accept this manuscript in its current form without significant revisions. 

As indicated in my comments in my previous review, this manuscript covers similar topics as a review published by the same authors in 2020 (https://doi.org/10.3389/fimmu.2020.00761), and doesn't significantly expand upon this review to identify a gap in the current knowledge.